# Facial Skin Microbiome Composition and Functional Shift with Aging

**DOI:** 10.3390/microorganisms12051021

**Published:** 2024-05-18

**Authors:** Allison Garlet, Valerie Andre-Frei, Nicolas Del Bene, Hunter James Cameron, Anita Samuga, Vimal Rawat, Philipp Ternes, Sabrina Leoty-Okombi

**Affiliations:** 1BASF Corporation, 540 White Plains Road, Tarrytown, NY 10591, USA; allison.garlet@basf.com (A.G.); nicolas.delbene@basf.com (N.D.B.); 2BASF Beauty Care Solutions, 32 Rue Saint Jean de Dieu, 69007 Lyon, France; valerie.andre-frei@basf.com; 3BASF Corporation, 26 Davis Dr, Raleigh-Durham, NC 27709, USA; hunter.james.cameron@basf.com (H.J.C.); anita.samuga@basf.com (A.S.); 4BASF SE, Speyerer Str. 2, 67117 Limburgerhof, Germany; vimal.rawat@basf.com; 5BASF Metabolome Solutions GmbH, Tegeler Weg 33, 10589 Berlin, Germany; philipp.ternes@basf.com

**Keywords:** skin aging, skin microbiome, gene ontology, diversity, *L. crispatus*, *C. acnes*, *C. kroppenstedtii*

## Abstract

The change in the skin microbiome as individuals age is only partially known. To provide a better understanding of the impact of aging, whole-genome sequencing analysis was performed on facial skin swabs of 100 healthy female Caucasian volunteers grouped by age and wrinkle grade. Volunteers’ metadata were collected through questionnaires and non-invasive biophysical measurements. A simple model and a biological statistical model were used to show the difference in skin microbiota composition between the two age groups. Taxonomic and non-metric multidimensional scaling analysis showed that the skin microbiome was more diverse in the older group (≥55 yo). There was also a significant decrease in Actinobacteria, namely in *Cutibacterium acnes*, and an increase in *Corynebacterium kroppenstedtii*. Some *Streptococcus* and *Staphylococcus* species belonging to the Firmicutes phylum and species belonging to the Proteobacteria phylum increased. In the 18–35 yo younger group, the microbiome was characterized by a significantly higher proportion of *Cutibacterium acnes* and *Lactobacillus*, most strikingly, *Lactobacillus crispatus*. The functional analysis using GO terms revealed that the young group has a higher significant expression of genes involved in biological and metabolic processes and in innate skin microbiome protection. The better comprehension of age-related impacts observed will later support the investigation of skin microbiome implications in antiaging protection.

## 1. Introduction

The skin environment is dynamic and alive with many commensal microorganisms that may contribute to maintaining healthy [1,2,3] and desirable skin [4]. These microorganisms form the skin microbiota, which is comprised of diverse bacteria, fungi, yeasts, viruses [5,6], archaea [7] and mites, principally Demodex [8].

The composition of skin microbial communities has been largely demonstrated to be dependent on the physiology of the skin site, with changes in the relative abundance of bacterial taxa associated with moist, dry and sebaceous environments, [9,10]. Sebaceous sites are dominated by lipophilic *Cutibacterium* species, while bacteria that thrive in humid environments such as *Staphylococcus* and *Corynebacterium* species, are preferentially abundant in moist areas. In contrast to bacterial communities, the fungal community composition is dominated by *Malassezia*. Interestingly, *Malassezia* populations are similar across core body site regardless of the skin physiology [11]. The skin is a barrier between internal human physiology and the surrounding environment and, as a result, the skin microbiota is also in constant contact with the external environment as UVs rays and pollution that can alter the skin’s microbiota composition [12,13,14]. In addition, it has been also observed that lifestyle habits like use of cosmetics [15] or diet [16] can also shape the skin microbial community.

Several studies have demonstrated that the skin microbiota composition and balance can change depending on internal and external factors such as skin integrity and physiological status [17], with various skin diseases such as atopic dermatitis [18], acne [19], psoriasis [20] and vitiligo [21]. Similarly, a shift in the cutaneous microbiota is also suspected in contributing to sensitive skin [22]. 

Studies on the cutaneous microbiome, including both microbial and genomic components, across different age groups have highlighted the dynamic nature of skin microbial communities: beginning at the early stage of life following the initial exposure to the maternal microbiome and continuing with shifts in community structure throughout puberty through to old age [23]. Thus, it has been shown that skin microbiome diversity increases with skin aging [24,25,26,27,28] and that this change in diversity is largely contributed by shifting Actinobacteria an Proteobacteria populations [24,29,30] with the *Cutibacterium* population decreasing on aged skin [26] and more particularly the *C. acnes* population [24,25,27,31,32], while *Corynebacteria* populations increase [24,26,29,32,33].

Skin aging has become a recurring concern even for younger people, mainly resulting from an increase in life expectancy [34]. In the antiaging skin care product panoply, anti-wrinkle products are the most requested. In that category, interest increased for products addressing skin microbiota supported by studies on differences by age, ethnicity, gender or environmental stress exposure. Consequently, correlations between the age-related changes in the skin microbiome and host metadata (physiology, medication, lifestyle, health or clinical skin features) were broadly investigated [24,25,26,27,28,30,31,32,33]. Interestingly, those correlations highlighted the link between specific taxa or even more precisely specific clades (for *S. epidermidis* or *C. acnes)*, and clinical skin characteristics such as high or low levels of skin moisture, transepidermal water loss (TEWL), porphyrin, sebum production, dullness, dark spots, wrinkle grade and dermal collagen quantity/quality. 

Moreover, changes in the microbial functional profile with aging were also reported thanks to deeper functional genomic analyses [25,27,32,35]. Finally, functional evaluation after keratinocytes and fibroblasts treatment by aging-modulated taxa, namely *Streptococcus pneumoniae* or *infantis* secretomes [35], or by *C. anes* and *Moraxella osloensis* [31], helped to better understand the contribution of those specific taxa to skin aging traits. 

Although the skin microbiome balance is now considered to be important for skin health, and that it is assumed that, on healthy people, it exhibits different characteristics according to various factors such as ethnicity, age, gender, and location of residence [30], the relationship between the skin microbiome related to the quality of aging skin still needs more exploratory studies. In order to develop holistic antiaging solutions, we sought to understand the composition of skin microbiota between young and aged skin, with a particular focus on the wrinkled area, using the most advanced techniques of DNA extraction and sequencing to have access to microbial composition with species-level resolution and functional genomic data. Our aim was to identify the bacterial changes in the specific wrinkled area that could not only contribute to bacterial homeostasis maintenance but also negatively or positively contribute to skin aging attributes, particularly inflammation, pH changes, wrinkles and skin mechanical properties, to later propose solutions to help improve skin aging signs. 

## 2. Materials and Methods

### 2.1. Subject Recruitment and Sample Preparation

Ethical statement: Study participants from the New York metropolitan area (healthy, aged 18–85 years) were recruited by BASF at the Tarrytown Consumer Testing Center (Tarrytown, NY, USA). The protocol for the present study (# TC-1216-004-074) was reviewed and approved by the BASF Tarrytown Consumer Testing Center Institutional Review Board (TCTC IRB). The protocol also covers the collection, sequencing, and data analysis of microbiome samples. Informed consent was obtained from all participating volunteers, and the experiments conformed to the principles set out in the WMA Declaration of Helsinki and the Department of Health and Human Services Belmont Report.

A total of 100 healthy female Caucasian volunteers aged 18–85 years were recruited. They were divided into two groups, a young and an old group following these criteria:

The young group was made of 50 subjects aged 18–35 years with crow’s feet wrinkles grade 0–1.

The old group was made of 50 subjects aged above 55 years with crow’s feet wrinkles grade 5–6.

All the volunteers gave their informed consent before participating in this study. Any and all identifying information was treated as protected health information. Protected health information (PHI) is considered under HIPAA and includes any individually identifiable health information.

Subjects were told to observe a 2-day wash-out period prior to the microbial sampling visit. They were told to avoid shampooing, stop using all topical products and wash their face only with water.

On the sampling day, participants were not permitted to wear any make-up or put on any facial products such as skin care products like moisturizer on their face. 

### 2.2. Skin Characterization of the Panelists

Clinical scientists evaluated the overall skin condition of the face, including erythema, psoriasis, sunburn, cuts, scar tissue, abrasions, lesions or other abnormal conditions or skin disease. Clinical scientists also conducted grading of the crow’s feet wrinkles. For both groups, wrinkle grade was checked using the Bazin atlas with the scale for crow’s feet wrinkles grade going from 0 to 6 [36]. In the study room, the environment was maintained at 20–30 °C and 45–65% of relative humidity without condensation.

#### 2.2.1. Questionnaire Acquisition Methodology

Under control of a technician at the test center, panelists completed a questionnaire that recorded medications, hygienic habits, dietary preferences, along with other lifestyle questions (closed questions or open-ended questions).

#### 2.2.2. Skin Hydration Measurement

Skin hydration was measured using a Corneometer (Courage + Khazaka electronic GmBH, Cologne, Germany) and expressed as arbitrary units (AU). The Corneometer measures the capacitance of the skin over a 10–20 µm thickness of the stratum corneum, which correlates to the water content.

#### 2.2.3. Skin pH Measurement

Facial pH was measured with a skin pH meter (PH 905, Courage +Khazaka electronic GmbH).

### 2.3. Bacterial and Fungal Sampling and DNA Extraction

A reproducible swabbing protocol was performed. Microbial samples were collected from either left or right side. For the younger group, sampling was conducted on the side with the lower wrinkle grade (grade 0 or 1), and on the side with the higher wrinkle grade (grade 5 or 6) for the older group. Microbiome sampling was performed using SCF-1 buffer (50 mM Tris buffer [pH 7.6], 1 mM EDTA [pH 8.0], and 0.5% Tween20 from Tecknova, Hollister, CA, USA) and HydraFlock swabs from Puritan Medical Products Guilford, USA supplied byVWR Dallas, TX, USA. The microbial samples were collected as follows:Inside the crow’s feet wrinkles (wrinkle), three samples were taken per wrinkle zone and collected for 30 s each using Puritan HydraFlock swabs mini-tipped. Following collection, all 3 samples were pooled prior to DNA extraction.Around the crow’s feet/under eye area, two samples were taken and collected for 60 s each using Puritan HydraFlock swabs with elongated tips.Following collection, both samples were pooled prior to DNA extraction.Finally, on the cheek adjacent to earlobe area (control), two samples were taken and collected for 60 s each using Puritan HydraFlock swabs with elongated tips. Following collection, both samples were pooled prior to DNA extraction.

Once the entire area was swabbed thoroughly, the swabs were immediately inserted into tubes containing the SCF-1 buffer and swirled to release specimens for at least 10 s to ensure the transfer of microorganisms into the solution. Tubes containing specimens were placed over ice. Within approximately 2 h, the samples were processed for DNA extraction.

The Epicentre^®^ MasterPure^TM^ kit (Epicentre, Madison, WI, USA) was used for DNA extraction. The purification kit protocol was followed with an exception to Step 1. Proteinase K was added directly to microcentrifuge tubes containing specimens in Cell Lysis Solution. Following Epicentre^®^ MasterPure^TM^ kit completion, removal of host DNA was achieved using a NEBNext^®^ Microbiome DNA Enrichment Kit (Epicentre, Madison, WI, USA).

### 2.4. Library Preparation and Sequencing

Whole-metagenome sequencing libraries were prepared from 26 µL of DNA using the NEBNext^®^ Ultra™ ll FS DNA Library Prep kit (Epicentre, Madison, WI, USA) DNA was first enzymatically fragmented followed by ligation of adaptors to the ends of the fragmented DNA. DNA was then purified, and size selected to remove excess adaptors and adaptor dimers using Ampure XP beads (Beckman Coulter, Indianapolis, IN, USA). Adaptor ligated DNA was then subjected to PCR amplification (17 cycles) with universal primer and an index barcode that is unique for each sample. The PCR product was cleaned, and size selected with beads to obtain a 250–600-base-pairs library. The library was then checked for quality and quantity on the Bioanalyzer. Libraries of more than 2 nM were submitted to paired-end (2 × 100 base pairs) sequencing on the HiSeq 3000 (San Diego, CA, USA). 

A total of 242 samples were sequenced over the 288 sampled. Samples were pooled together in batches of 6 for every lane of sequencing, after which the data were de-multiplexed, checked for quality and handed off for bioinformatics and data analysis. 

### 2.5. Analysis of the Skin Microbiome

#### 2.5.1. Taxa and Statistical Analysis

Contaminants, adaptors, and known artifacts were removed from the raw reads using bbduk (https://sourceforge.net/projects/bbmap/ (accessed on 9 January 2019)). Quality trimming was applied to the ends of reads targeting q10 using bbduk, and reads shorter than 70 base pairs after trimming were removed. Human reads were removed from the dataset by mapping reads using Bowtie2—sensitive to hg38 [37]. Read pairs in which both reads mapped concordantly to hg38 were removed. All other reads were processed using MetaPhlAn2 to generate a taxonomy abundance profile for each sample [38].

For the simple data analysis model, non-metric multidimensional scaling (NMDS) was used to evaluate the results. The Bray–Curtis dissimilarity matrix derived from the microbiome composition was used to construct clusters of samples. Non-metric multidimensional scaling (NMDS) was used to evaluate the sample structure. The statistical significance of the association between microbiome composition and cohort was determined using one-way/two-way ANOVA for multiple comparisons, the *t*-test, and the Chi-square test, as appropriate; differences *p* < 0.05 were noted as significant. The false discovery rate (FDR) was also reported with the Benjamini–Hochberg procedure. The spatial distribution of the species with significant changes in the relative abundance between the groups in the different skin sites was generated. All analyses were performed in R with different packages.

#### 2.5.2. Metadata Analysis

Metadata were either captured in the questionnaire or were part of the recorded skin measurements (hydration and pH). These values subsequently form the input variables for the downstream redundancy analysis (RDA) and linear model, looking at whether input variable changes could explain changes in the output variables (genomic reads binned per species). When there was a high correlation between two input variables, then these variables were removed from downstream analysis.

A stepwise procedure was performed to add terms to a model, which includes as many significant terms as possible, while also retaining a good fit to the model. This was performed in a stepwise process called constrained ordination. Analysis started with a model that only includes the term that explains the highest variance fraction. Then, the next term was added (and so on), until the model R2 does not increase any further. 

#### 2.5.3. Taxonomy Analysis—The Biological Model 

The purpose of this metadata analysis was first to determine which metadata could potentially be correlated with each other. If two metadata were correlated, only one of two was introduced in the model to avoid biasing the biological analysis. The goal was to limit model terms to just those that were descriptive on their own.

A filtering step was first conducted to remove bacteria. Gene ontology with less than a 25% prevalence in the total dataset was removed. A second threshold was set up also to remove species that show very low abundance across samples; this threshold is typically set close to what we consider our detection threshold 10^−6^. Terms were normalized using quantile normalization and log10 transformation. The lifestyle data were utilized to correlate variables between subjects and ultimately built into the biological model (Figure 1).

This model only considers the terms found significant using the RDA ANOVA test (at the *p*-value < 0.01 threshold). Both pH and Corneometer values were not found to add to the model goodness of fit; therefore, they were not included.

#### 2.5.4. Functional Analysis

Relative abundance profiles from each sample were merged into an experiment-wide profile and this was used to generate the nucleotide database used in HUMAnN2 v0.11.1 [39]. The database was generated for the whole dataset rather than for specific samples to create a representative skin microbiome database that better captures interpersonal variability. HUMAnN2 was run on each sample using the custom nucleotide database and the UNIREF90 database (April–October 2017) clustered at 90% identity available in HUMAnN2.

## 3. Results

### 3.1. The Skin Characterization of the Panelists

A total of 95 panelists completed this study: 49 in the older group and 46 in the younger group. Age, skin pH, and Corneometer readings for hydration were taken for all participants during this study. Results are presented in Table 1. 

Statistical comparisons did not show a significant difference between the two groups for Corneometer and skin pH values.

### 3.2. Library Preparation Results

Low input biomass and high backgrounds of host DNA make library preparation for NGS sequencing a challenge. A total of 209 samples had 2 × 100 base pairs successful library preparation out of a total of 288 samples. Among those 209 successful libraries, complete wrinkle, crow’s feet, and control area samples were 25 and 21 for young and old groups, respectively. All libraries were sent for HiSeq sequencing.

The majority of reads passed quality control analysis (~98%) but approximately 80% of these reads were human. These human reads were subsequently removed and the remaining 20% of microbial reads were analyzed for taxonomy and gene ontology.

### 3.3. Comparison of the Microbiome of Young and Old Skin

#### 3.3.1. Taxonomy Results—Group Skin Microbiome Composition and Structure

Taxonomy results are an indication of the taxa relative abundance between groups. A simple model of the data highlights the differences between the old and young groups and the differences of microorganisms at different sample sites. There are 52 significant (*p* < 0.05) microbial differences between the old and young groups. The full set of taxonomic data can be found in Appendix A.

The average relative abundance of species was visualized using krona plots. Considering all three zones, clear shifts can be seen in Actinobacteria (47% in old and 67% of all microorganisms in the young group), Proteobacteria and Firmicutes (18% in old and 10% of all microorganisms in young), depicted in pink, green and brown color, respectively. In the older group compared to the younger group, there is a decrease in Actinobacteria, an increase in Firmicutes and Proteobacteria (Figure 2a,b). 

Furthermore, in Actinobacteria, when comparing the *Propionibacteriacea* and *Corynebacteriea*, there is a clear shift when comparing the old and young groups. *Propionibacteriacea* have higher relative abundance in the young group, which appears be reduced in the old group; the most striking difference is observed with *Cutibacterium acnes* (formerly known as *Propionibacterium acnes*). For *Corynebacteriacea*, *Corynebacterium kroppenstedtii* was the most shifted with a significant higher relative abundance in the old group compared to the young group.

For Firmicutes, in the old group, we observed an increase in some *Staphylococcus* and *Streptococcus* species. In the young group, we observed a higher relative abundance of lactic acid bacteria, namely within the *Lactobacillus* species. *Lactobacillus crispatus* is the most represented in the young group (Figure 2c, d). It represented 0.3% of all microbial reads, while it is 0.007% of all microbial reads within the old group.

Additionally, the Proteobacteria phylum increased from 3% to 9% of all microbial reads in the old group (Appendix A). We also observe that there are higher relative abundances of opportunistic pathogenic bacteria such as *Escherichia* and *Pseudomonas* species.

Both groups’ microbiota were characterized by their alpha diversity. Alpha diversity index comparison highlighted a significant higher (*p* = 4.35 × 10^−7^) alpha diversity in the old group when compared to the young group using the Kruskal–Wallis test (Figure 3).

Non-metric multidimensional scaling (NMDS) normalized to the highest variables in captured metadata analysis showed that the younger group appears to cluster more readily together, indicating more similarities. Conversely, the older group is spread, indicating more differences (Figure 4).

#### 3.3.2. Taxonomy Results—Prevalence Analysis

Another comparison was based on species prevalence, which was calculated based on presence or absence observations between groups. The full set of taxonomic data can be found in Appendix A.

Figure 5 highlights organisms with interesting disparities in prevalence between the old and young groups, namely *L. crispatus* and *iners*, *S. mutans* and *gordonii*, *C. kroppenstedtii* and *massiliense*, and *Betapapillomavirus 1*. 

Prevalence analysis particularly revealed a higher presence of lactic acid bacteria in all zones of the young group and a decrease in the old group (Figure 6a). Among the most prevalent lactic bacteria, most of them were *Lactobacilli*. The most extreme disparities in prevalence were for *L. iners* and *crispatus*, which are much less present on the skin from the older group.

Analysis of the three zone compositions showed that *L. crispatus* occurrence was the most decreased. If we look specifically at the wrinkle zone, the highest decrease was observed with *L. crispatus*, which is below the detection limit within the wrinkle zone of the older group.

Regarding relative abundance (Figure 6b), among the four more abundant lactic acid bacteria present, *L. crispatus* was one of the most decreased, especially in the crow’s feet zone, and was not detected in the wrinkle hollow part of older skins.

#### 3.3.3. Metadata Analysis Results with the Biological Model

Metadata were captured either in the questionnaire data, sample location, skin pH, or hydration measurements. Significant data are presented in Table 2.

The most highly correlated factors in metadata analysis were body weight and BMI (body mass index) and shampoo frequency and conditioner frequency, so only one over two was kept for metadata analysis for the biological model. Redundancy analysis significant terms model with the lowest *p*-value are shown in Table 2 below. These metadata were used to refine a biological model.

#### 3.3.4. Taxonomy Results—The Biological Model

The biological model uses terms (captured either in the questionnaire data, sample location, skin pH, or hydration measurements) that significantly influence skin genomic read information, when grouped into species. The following terms were used as independent variables in the estimation of effect linked to the relative abundance of species in each group. Volunteers’ characteristics and skin parameters like group age group (old or young), skin pH, Corneometer (skin hydration), and sample location (control, wrinkle, or crow’s feet) were used as well as lifestyle factors like hormone supplements, shower frequency, average sleep, sleep pattern changes, processed foods frequency, body mass index, tobacco use frequency, antiaging products frequency, cleansing conditioner frequency, body wash (non-medicated) frequency and personal birth.

The model effect estimate together with the intra-group variability were considered by the model to attach a statistical significance (*p*-value).

Positive X-values denote higher species effect and relative abundance in the younger group while negative X-values denote higher species effect and abundance in the older group. Increasing Y-values indicate a more significant effect.

In the biological model (Figure 7), we saw *C. acnes*, *P. nanceiensis*, and *L. crispatus* have the highest significant effects in the young group while *C. kroppenstedtii*, *R. dentocarlosa*, *C. pseudogenitalium*, *S. gordonii*, C2likevirus (Lactococcus phage), *Betapapillomavirus* 1, *S. mutans*, and *S. vestibularis* had the highest effects in the old group.

The age group in which the sample belongs (Figure 7) and the personal birth mode (Appendix A) are the two variables with the highest significant influence on the presence of *L. crispatus* in the sample in this model.

#### 3.3.5. Functional Genomic Results

Sequences were analyzed and compared between groups. Due to the low microbial sequence (~20%) and the high relative abundance of *C. acnes* (74% and 34% average relative abundance in the young and old group, respectively), most of the genes mapped back to the *C. acnes* reference database. Nevertheless, there are some interesting indications in the functional genomic data that are supported by some of the taxonomy conclusions. Functional genomic results were analyzed by investigating differences in gene ontologies (GO).

Microbial functional genes are described using gene ontologies (GO terms). GO terms were analyzed between the old and young groups. In the old group, significant GO terms are far fewer than for the young group with significant terms totaling 3 and 160, respectively. The GO terms related to the old group mainly reflect ontologies related to aerobic respiration (GO 0032501 organismal process, GO 0016469 proton transporting two sector of ATPase complex and GO 0006119 oxidative phosphorylation), which may suggest a shift from anaerobic metabolism in a larger *C. acnes* population. In addition to that, their effect size or magnitude of change was less important compared to the significant GO terms in the young group.

The top 20 GO terms in the young group with the largest effect in the young group seems related to metabolism (Biological process and molecular function) (Figure 8). Furthermore, among the 160 GO terms, we do see other GO terms in the young group (Table 3), which may contribute to low microbial diversity (GO0031640 killing of cells of other organism, GO0001871 pattern binding, and GO0009372 quorum sensing) and potential mechanisms for antiaging (GO0016209 antioxidant activity).

## 4. Discussion

While skin aging is not a disease state of skin, it is an inescapable process that affects all human beings. The importance of aging lies in the enormous consumer demand for products that can prevent or improve its signs like wrinkles [40]. Interestingly, in an attempt to predict a person’s age from a microbiome sample from different body sites and to link skin age and microbiota, Huang and coworkers have combined several large studies from different countries to determine which body site’s microbiome could predict human age; intriguingly, the skin microbiome was the best in predicting volunteers chronological age [41]. 

To better understand the shift in skin microbiota on aged skin, and further its role in skin protection against aging, we conducted a study to discover differences in microbiota structure between an old group and a young group, particularly focusing on the wrinkled area of the skin. To this end, we used whole-genome sequencing (WGS) to gain deeper insight into the composition and to gain access to the microbial structure at a species level as well as microbial functional potential.

The results of this study confirmed some observations previously made using 16S rRNA sequencing studies [24,25,26,28,30,33,42] or using more recent whole-genome sequencing (WGS) studies [27,31,32]. The study results first confirmed that microbial diversity increases during aging [24,25,26,27,28,29,41], largely contributed by shifting Proteobacteria populations and Actinobacteria [24,28,29,30] or Lactobacillales [43]. As reported previously on forehead and/or cheeks, we also observed that in older facial wrinkled skin, there is a decrease in *Cutibacterium*, formerly *Propionibacterium* [24,25,26,27,29,31,32] and an increase in *Corynebacteria* [24,29,33], namely a decrease in *C. acnes* [24,25,27,31,32] and an increase in *C. kroppenstedtii* [32,33]. 

Contrary to previously published studies using 16S rRNA sequencing, the use of WGS allowed us to obtain more detailed information with respect to the composition of the microbiome of young and old skin at the species level. Moreover, we were also able to perform a functional genomic study.

Using WGS, we observed that the shift in *Corynebacteria* was more complex than previously reported by Shibagaki, Jugé, Zhou, Li and coworkers [24,29,32,42]. This complexity has been also partially highlighted in a study conducted by 16S rRNA on 495 North American subjects with some species increasing like *C. kroppenstedtii*, while others staying constant or decreased [33]. Another interesting observation is that many of the bacteria found to be significantly higher in the wrinkle zone are members of the oral microbiome niche. We found 13 out of the 19 significant wrinkle bacteria also inhabit the oral microbiome. This was also observed in a skin microbiome aging study published by Shibagaki in 2017 [24]. 

These diversity shifts between groups suggest a shift from a lipophile-dominated ecosystem of young skin to a less lipophilic and more varied metabolic potential on older skin. The key species within the young skin microbiota appears to be *C. acnes*, which is also considered as a commensal microorganism. Dominant particularly in oily skin sites in younger adults, it has key roles in immunomodulation, epithelial barrier maintenance and in protecting the host from pathogens [27]. *C. acnes* metabolizes triglycerides to produce free fatty acids with a broad-spectrum inhibitory effect preserving the microbiome balance of young skin [44,45]. *Cutibacterium* may also modulate the immune responses and suppress inflammation to further slowdown the aging process by altering conjugated linoleic acid generation and influencing the expression of antimicrobial proteins by *Staphylococcus* [42]. Its decrease in older people may consequently make aged skin more sensitive to pathogen colonization, Staphylococci infection and disease risk [27]. 

Regarding presence and absence analysis, the prevalence of the most abundant species was similar in both groups. However, some prevalence disparities between the two groups have been demonstrated. The old group is characterized by a higher presence of *S. gordonii*, *S. mutans*, *C. kroppenstedtii* and *C. massiliense*.

A study conducted in China by Li and coworkers showed that in the older group, the higher relative abundance of Streptococcus correlated with exacerbation of the progress of emerging hyperpigmentation (brown) spots, wrinkles, texture, and porphyrin expression, leading them to consider these species as accelerating factors for skin aging [42]. However, Kim et al. demonstrated that *S. infantis* or *pneumoniae* dominate on the skin of young Korean women and produce spermidine, increasing the expression of genes related to extracellular matrix synthesis in fibroblasts (collagen, elastin, fibrillin 1) or skin barrier (filaggrin, ABCA12) in keratinocytes [35]. Deeper analysis of Streptococcus genus variation may be thus useful to better understand functional impacts on skin aging.

The dominance of *C. kroppenstedtii* on older skin can be driven by the metabolism and production of free fatty acids, which have been found on older skin [46]. Although considered as a skin commensal, there is evidence that this species can act as an opportunistic pathogen [47]. It was among the most abundant bacteria on skin of patients suffering from rosacea and redness compared to healthy skin [48,49]. To our knowledge, the presence of *C. massiliense,* as a member of the skin microbiome, has never been previously reported.

In this study, the skin of the young group is also characterized by a higher presence of *Lactobacilli*, with the highest prevalence in *Lactobacillus iners* and *crispatus*. The Lactobacilliale order has been described to be present on the skin of Finnish children aged one year or less and it has been reported that their populations decrease in favor of Actinobacteria and Proteobacteria [43]. Lactobacillus dominance in young Caucasian women was also confirmed by Howard and coworkers [26]. This occurrence on the skin confirms that they are seeded on the skin after delivery through the birth canal, as these bacteria can be found originating from the vagina [50].

The *Lactobacillus* genus has been detected on face skin of middle-aged Chinese and young females in China [42,51]. *Lactobacillus* species presence on skin could be explained by their capacity to bind some skin proteins such as keratin [52]. *Lactobacilli* have the potential beneficial role in the skin habitat, where they can exert multifactorial local mechanisms of action against pathogens and inflammation [53]. Even if not a dominant skin microbiota member in terms of relative abundance, their presence may develop protective effects from aging for several reasons. *Lactobacillus* can produce various antimicrobial substances and induce anti-inflammatory Treg cells to reduce inflammatory injury caused by UV radiation [54]. Furthermore, *Lactobacilli* has been proven to have antioxidant properties in vitro [55]. All these valuable *Lactobacillus* properties can potentially improve conditions for the decrease in collagen synthesis and wrinkle formation [56]. Moreover, treatment and prevention of atopic dermatitis using *Lactobacillus*, among 13 described species, have been published [57]. Interestingly, *L. crispatus* probiotic was shown to improve skin barrier damage and inflammation through the production of tryptophan metabolites that act as aryl carbon receptor agonists [58]. When associated with *Lacticaseibacillus paracasei* it was shown to improve seborrheic dermatitis symptoms [59]. Additionally, *Lactobacillus* genus also acts synergistically with Staphylococcus, which likely leads to the enrichment in the biosynthesis of antibiotics could promote the formation of mature and complete immune barriers or host defense, to further protect during photoaging [42].

In the biological model, *L. crispatus* has the highest effect on the skin of young group skin (Figure 7). Regarding *L. crispatus* importance in the young group, our results support the results of a study conducted to compare the microbiome and mycobiome of Korean women of different ages [25]. We hypothesize that a larger *C. acnes* population on young skin helps skin acidification and supports environmental conditions that promote *Lactobacilli* and *L. crispatus* populations on young skin.

*Lactobacillus crispatus* and *Lactobacillus iners* have been reported to be present on young female Korean skin [25]. *L. crispatus* is a well-known microorganism associated with a healthy vaginal microflora. *L. crispatus* is a homofermentative organism which undergoes substrate level phosphorylation producing lactate from glucose; it can produce bacteriocins and other metabolites like organic acids to protect skin from pathogenic microorganism development [60,61]. Metadata suggest that there is a significant positive correlation with vaginal birth and the presence of *L. crispatus* on skin. This indicates that *L. crispatus* can be seeded on skin from healthy mothers during birth. Furthermore, *L. crispatus* have some multifunctional proteins, which can adhere to host tissue. It has been shown that within an acidic pH condition, a *L. crispatus* strain ST1 was able to bind a reconstituted basement membrane preparation [62]. These multifunctional proteins can be released under stress conditions or in the presence of antimicrobial LL-37 produced by human skin cells and interact with skin extracellular matrix proteins, which may support different skin microbiome interactions during skin aging. Finally, *L. crispatus* has surface proteins (S-layer proteins) which bind to some extracellular matrix proteins like collagen and fibronectin [63]. Theses structural characteristic of *L. crispatus* may suggest the capacity to interact with skin and its components.

Considering the functional genomic data, we can find many lipases including triacylglycerol lipase, lysophospholipase, and phospholipase which are not significantly different between the groups. These results could be misleading, due to the resolution of the data, i.e., only 20% of total sequence was microbial and the largest amount sequence is from the high population of *C. acnes* in both groups (74% and 34% average relative abundance in the young and old groups, respectively). The major nutrients source on the skin is sebum which consists in triacylglycerides, cholesterol, squalene, and wax esters. There is also a high concentration of free fatty acids on skin, which is produced by lipase-mediated hydrolysis of triacylglycerides from resident microorganisms [64,65]. This hydrolysis produces both free fatty acids and glycerol, which can then be utilized by the microbial community. However, minor species on young skin like *Lactobacilli* can benefit from major species like *C. acnes* which acidify the skin to survive and express some genes such as “killing activities” and “quorum sensing”, allowing them to compete against other microorganisms.

Finally, we have shown a functional genomic difference between young and old group. The GO terms analysis reported genes involved in enzymatic activities and metabolic processing of sugar, proteins and lipids suggesting a higher metabolic activity in the young skin microbiome. This is consistent with previous studies showing that pathways involved in energy metabolism by bacteria (such as glycolysis/gluconeogenesis, citrate cycle, pentose phosphate pathway, fructose and mannose metabolism, galactose metabolism, d-alanine metabolism, and thiamine metabolism) were predominant in the young group, whereas degradation-related pathways and lipopolysaccharide biosynthesis pathway predominated in the old group [25]. Zhou et al. also reported a richer but not more abundant antibiotic resistance reservoir in the facial microbiome of older women that correlated for instance the abundance of fusidic acid resistance genes with biophysical parameters such as the decrease in dermal water content [32].

## 5. Conclusions

Within this study, we were able to observe significant differences between the skin microbiome and the pan genome of the old and young groups.

The results of this study allow us to confirm certain observations made previously by other studies, namely that there is an increase in the cutaneous microbiome diversity with age and that *C. acnes* and lactobacillus populations, especially *L. iners and crispatus*, decreased with age, whereas a higher prevalence and abundance of *C. kroppenstedtii* and some bacteria of the Streptococcus genus were observed. 

This study contributes to a better understanding of the skin microbiota features of young and older skin and their probable function on the skin. However, this study showed some limitations in the genomic and functional analysis that was not able to completely explain the population dynamic shifts observed between the two groups. The major reason for this is that most of the sequences were related to the *C. acnes* population due to its high proportion on skin as well as the high amount of contaminant human DNA that had to be removed. Overall, the advantage of using whole-genome sequencing remains in the high-resolution sequencing at the species level and the discovery of the role of minority species in skin aging. With this high-resolution sequencing technique, we can begin to understand the influence of minority species on the skin microbiome and interrogate their role in skin aging.

The next steps are to further elucidate the effective functions of some microbial targets related to young skin such as *Lactobacilli*. This could bring some opportunities to replenish the skin with these strains to help fight skin aging phenotypes such as inflammaging and skin stress.

## Figures and Tables

**Figure 1 microorganisms-12-01021-f001:**
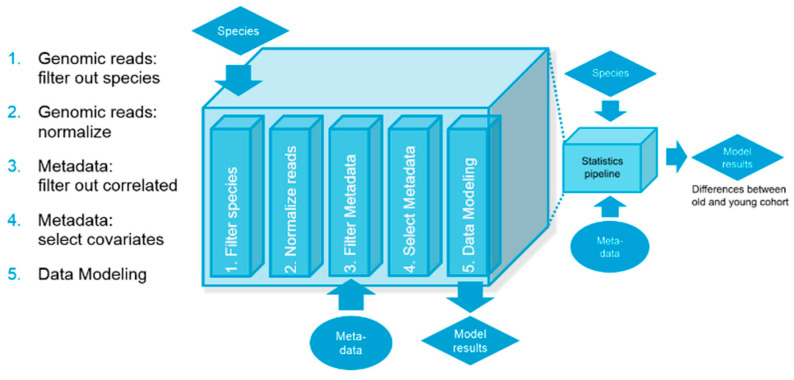
Schematic on the biological model for relative abundance.

**Figure 2 microorganisms-12-01021-f002:**
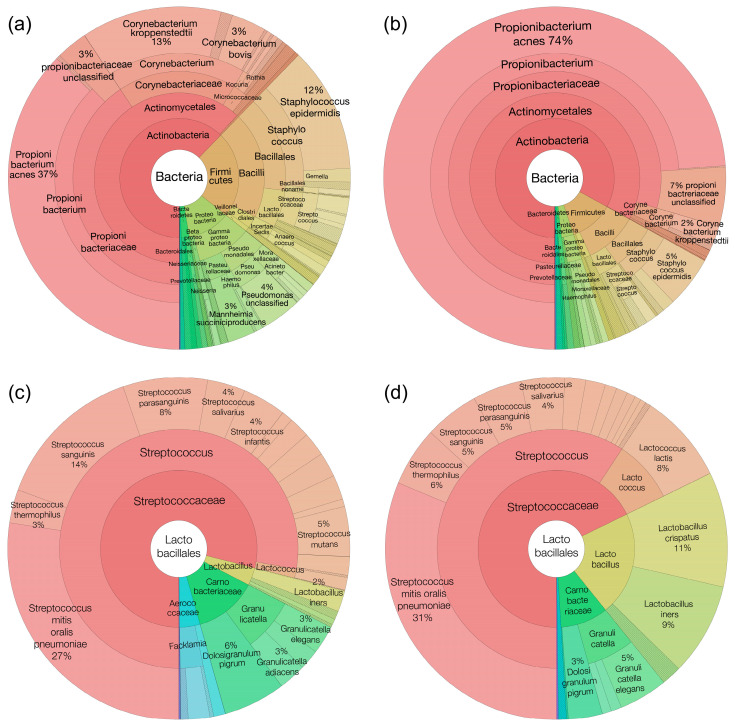
Krona plots highlighting the old and young skin sample bacterial composition. Actinobacteria, Proteobacteria and Firmicutes composition in the old group (**a**) and the young group (**b**). Lactobacillales composition in the old group (**c**) and the young group (**d**).

**Figure 3 microorganisms-12-01021-f003:**
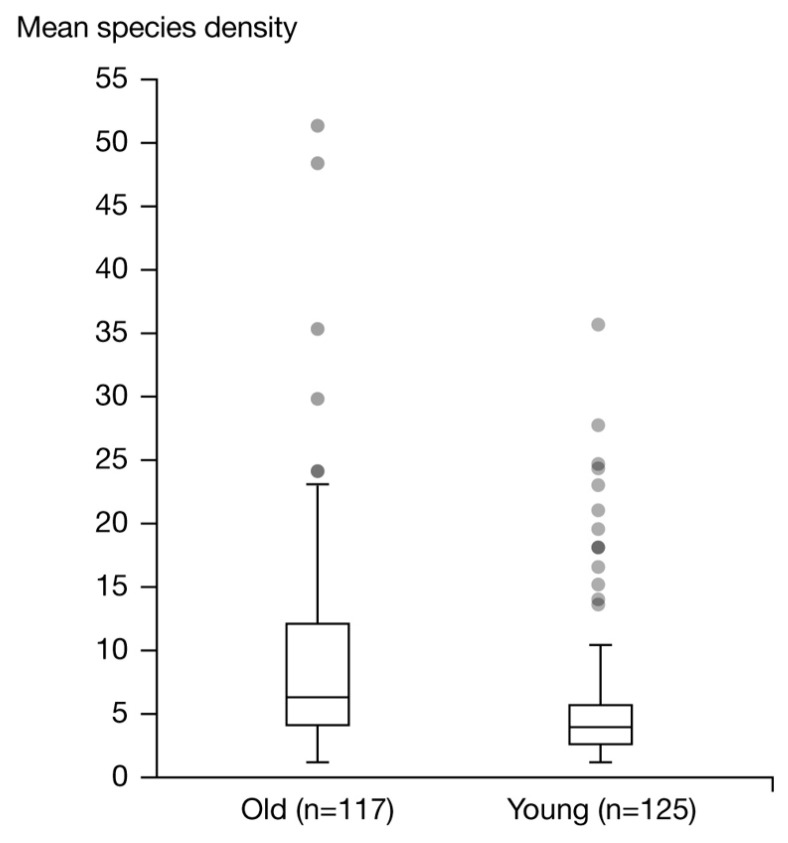
Alpha diversity on a species-level in the older and young groups.

**Figure 4 microorganisms-12-01021-f004:**
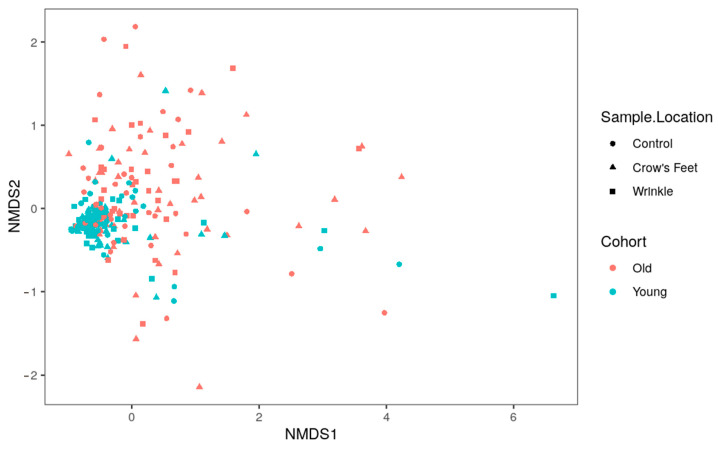
Non-metric multidimensional scaling (NMDS) analysis of the old and young skin microbiome.

**Figure 5 microorganisms-12-01021-f005:**
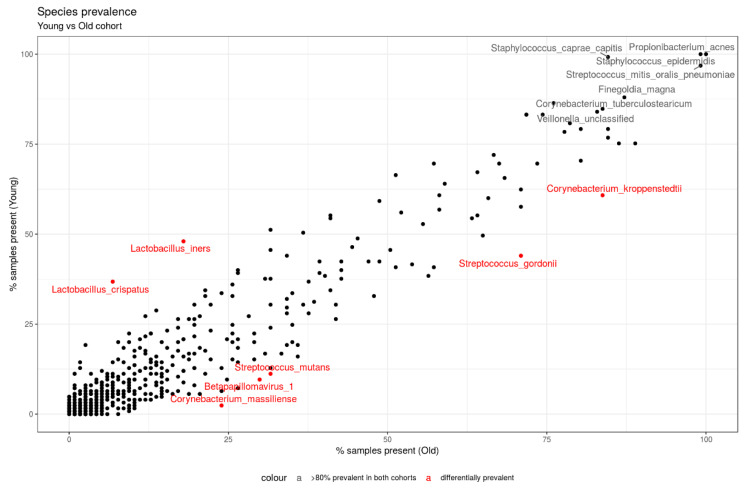
Scatter plot highlighting the percentage of samples present in the young group (y axis) and the percentage of samples present in the old group (x axis). Organisms highlighted in red present disparities in prevalence between the two age groups. Species with more than 80% prevalence in both groups are named in gray font. Any species that is less than 25% prevalent in any of the two groups is not represented.

**Figure 6 microorganisms-12-01021-f006:**
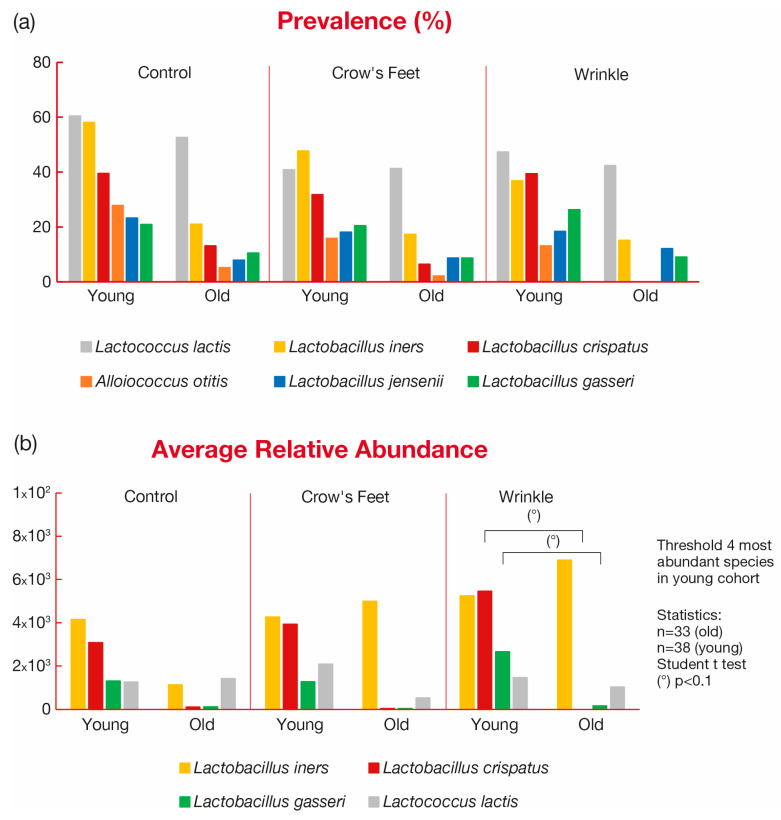
Variations in the most prevalent (**a**) and abundant (**b**) lactic acid bacteria.

**Figure 7 microorganisms-12-01021-f007:**
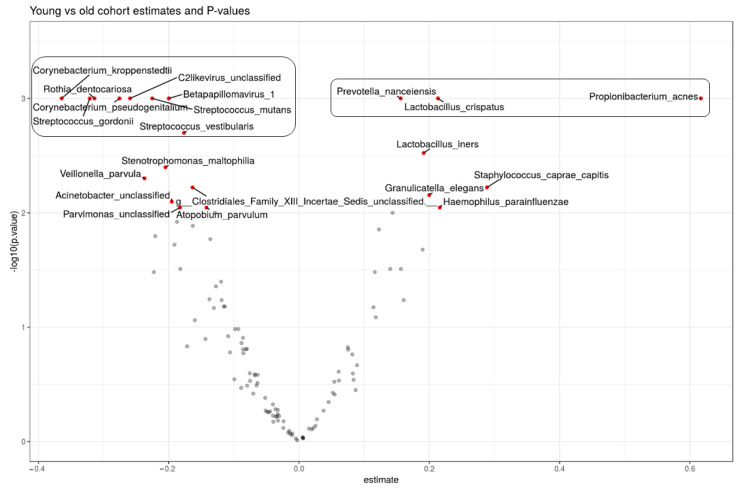
Volcano plot highlighting significant species in the biological model and effect size in the old and young groups. The Y axis depicts the *p* value, and the X axis depicts the effect size.

**Figure 8 microorganisms-12-01021-f008:**
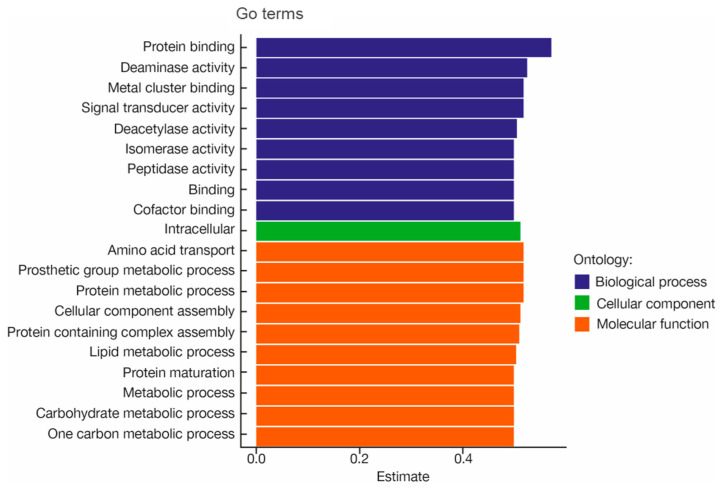
The top 20 GO terms in the young group in order of effect size. Significantly enriched GO terms were selected based on a *p* < 0.05. GO terms of the categories of biological processes, cellular components, and molecular functions are depicted in red, green, and blue, respectively.

**Table 1 microorganisms-12-01021-t001:** Subjects skin characteristics.

	Descriptive Statistics	Age	Skin pH	Corneometer
Old group	Average	62.82	5.05	46.10
	Median	62.00	5.06	43.00
	Standard deviation	4.69	0.59	11.37
	Subjects	49	N/A	N/A
Young group ^1^	Average	26.3	4.93	50.5
	Median	26.0	4.89	50.3
	Standard deviation	5.0	0.47	11.9
	Subjects	47	N/A	N/A

^1^ One subject from the younger group did not return for the rest of this study. N/A non-applicable.

**Table 2 microorganisms-12-01021-t002:** Significant metadata following redundancy analysis.

Metadata	Adjusted *R*^2^	AIC ^1^	F	*p*-Value
Hormone supplements Y.N.	0.045	734.046	6.950	0.002
Shower frequency	0.066	729.960	6.061	0.002
Average sleep (open-ended)	0.076	728.183	3.726	0.002
Sleep pattern changes Y.N.	0.086	726.856	3.265	0.002
Processed foods frequency	0.093	725.983	2.804	0.004
pH	0.099	725.353	2.554	0.006
BMI	0.105	724.822	2.447	0.006
Tobacco use frequency	0.110	724.320	2.408	0.008
Antiaging products frequency	0.116	723.673	2.537	0.004
Cleansing conditioner frequency	0.122	723.182	2.375	0.006
Body wash non-medicated frequency	0.128	722.510	2.538	0.002
Personal birth	0.133	722.103	2.275	0.008

^1^ AIC: Akaike’s criterion; Y.N.: Yes or No.

**Table 3 microorganisms-12-01021-t003:** Other GO terms with significant effect in the young group.

GO Terms in the Young Group	Variable	*p*-Value	Estimate
GO.0016209..antioxidant.activity	Young group	0.001	0.352
GO.0009372..quorum.sensing	Young group	0.001	0.376
GO.0001871..pattern.binding	Young group	0.005	0.251
GO.0031640..killing.of.cells.of.other.organism	Young group	0.04	0.212

## Data Availability

The experimental metagenomic sequence data have been published in Bioproject under the reference PRJNA1099173.

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
