# Peer review of "Facial Skin Microbiome Composition and Functional Shift with Aging"

_microorganisms, 2024, doi:10.3390/microorganisms12051021_

Round 1

Reviewer 1 Report

Comments and Suggestions for Authors

The manuscript titled ‘Facial skin microbiome composition and functional shift with  aging' offers an interesting point of view. However, a few changes would increase the value of the paper:

In the Discussion section, the authors need to explain more clearly the utility of the data obtained in medical practice.

The two groups analyzed in this study should be clearly defined in the Abstract.

The aim of the study must be clearly stated.

I consider it more appropriate to use "groups" instead of "cohorts".

The paragraph from lines 66-69 should be moved to the Discussion section.

Please clarify "avg sleep open enc" and "Y.N." in Table 2.    In the Material and Method Section there is no data regarding ethics committee approval.

Lines 359-364 include the aim of the study and should be moved from the Discussion section to the end of the Introduction section.

The names of bacteria should be written out in full when they first appear in the text and then should be abbreviated (e.g., S. mutans).

The conclusions are too long; some paragraphs can be moved to the Discussion section (lines 484-493).

English grammar should be revised throughout the article.

Comments on the Quality of English Language

English grammar should be revised throughout the article.

Author Response

1. Summary

Thank you very much for taking the time to review this manuscript and suggesting some changes in order to increase the value and the comprehension of this paper. Please find the detailed responses below and the corresponding revisions/corrections highlighted/in track changes in the re-submitted files.

2. Questions for General Evaluation

Reviewer’s Evaluation

Response and Revisions

Does the introduction provide sufficient background and include all relevant references?

Can be improved

Corresponding response in the point-by-point response letter. The same as below

Are all the cited references relevant to the research?

Yes

Is the research design appropriate?

Yes

Are the methods adequately described?

Yes

Are the results clearly presented?

Can be improved

Are the conclusions supported by the results?

Can be improved

3. Point-by-point response to Comments and Suggestions for Authors

Comments 1: [In the Discussion section, the authors need to explain more clearly the utility of the data obtained in medical practice]

Response 1: [Type your response here and mark your revisions in red] Thank you for pointing this out. We understand and agree that potential medical application is not clearly stated. Clearly, these results have applications in cosmetology, but they may also have applications in the medical field. The reduced abundance of Cutibacterium acnes, which is known to modulate the immune response and fight inflammation, make older skin more susceptible to infection and inflammatory skin diseases. Therefore, we add in the discussion section this potential application [page 14, Paragraph Discussion, line 461, 483 and 504].

Comments 2:

The two groups analyzed in this study should be clearly defined in the Abstract.

Response 2: We have, accordingly, described the two groups in the abstract indicating the age range of two groups [Page 1, lines 21 and 24]  

Comments 3: The aim of the study must be clearly stated.

Response 3: We agree that the description of the aim of the study can be improved. We therefore make it more explicit. [Page 2; lines 92 to 96].

Comments 4: I consider it more appropriate to use "groups" instead of "cohorts".

Response 4: We have replaced the term “cohort” by the term “group” in all the manuscript.

Comments 5: The paragraph from lines 66-69 should be moved to the Discussion section.

Response 5: We agree with this comment and move this paragraph to the discussion section, [Page 13 lines 418 to 423]

Comments 6: Please clarify "avg sleep open enc" and "Y.N." in Table 2. In the Material and Method Section there is no data regarding ethics committee approval.

Response 6: Thank you for raising this point about the abbreviation used in table 2 and the validation of this study by an ethics committee. We have thus updated the table 2, precise the abbreviation used [Page11; line 358]

For ethic committee approval, we add an ethical statement in the section “Subject recruitment and sample preparation” [Page 3; lines 100-124].

Comments 7: Lines 359-364 include the aim of the study and should be moved from the Discussion section to the end of the Introduction section.

Response 7: We agree with this proposal and move this to the end of introduction [page2 line 92-96]

Comments 8: The names of bacteria should be written out in full when they first appear in the text and then should be abbreviated (e.g., S. mutans).

Response 8: We totally agree with that and have changed this in all manuscript.

Comments 9: The conclusions are too long; some paragraphs can be moved to the Discussion section (lines 484-493).

Response 9: We found this proposal interesting, so we moved Lines 484-493 to line 441 page 14.

4. Response to Comments on the Quality of English Language

Point 1: English grammar should be revised throughout the article.

Response 1: the manuscript has been reviewed by a native English speaker (from UK)

5. Additional clarifications: No additional clarifications

Reviewer 2 Report

Comments and Suggestions for Authors

Review on “Facial skin microbiome composition and functional shift with aging” for manuscript ID microorganisms-2873879

In this manuscript the authors describe changes in microbial community of the women’s facial skin (for young and older groups separately). Whole genome sequencing data indeed valuable and could help to improve our knowledge about face skin microbiome, its diversity and evolution with aging.

In the Introduction section authors describe key studies about human age relationship and skin microbiome. Unfortunately, there are number of recent studies on the topic were missed:

Myers T, Bouslimani A, Huang S, Hansen ST, Clavaud C, Azouaoui A, Ott A, Gueniche A, Bouez C, Zheng Q, Aguilar L, Knight R, Moreau M and Song SJ (2024) A multi-study analysis enables identification of potential microbial features associated with skin aging signs. Front. Aging 4:1304705. doi: 10.3389/fragi.2023.1304705

Howard, B., Bascom, C. C., Hu, P., Binder, R. L., Fadayel, G., Huggins, T. G., et al. (2022). Aging-associated changes in the adult human skin microbiome and the host factors that affect skin microbiome composition. J. Investig. Dermatol 142 (7), 1934–1946.e21. doi:10.1016/j.jid.2021.11.029

Larson, P.J., Zhou, W., Santiago, A. et al. Associations of the skin, oral and gut microbiome with aging, frailty and infection risk reservoirs in older adults. Nat Aging 2, 941–955 (2022). https://doi.org/10.1038/s43587-022-00287-9

My questions and comments about Results and Discussion:

Section 3.1.3 is unclear at whole for non-specialists. Table 2 has no references in the main text, what the units of measured metadata and how the data were used to build a model? The description in 2.5.2 is very short, it could be clearer with the scheme of data analysis.

Figure 1: most of the text is unreadable, please correct the font or layout

L292: “highly correlated factors” – correlated with which variable?

L375: reference [36] devoted to the skin disease, not age-related

L379: here authors talk about bacterial species (13 of 19), not microorganisms?

Methods section comments:

First question – the raw sequencing data are missing, so the results cannot be reproduced. The DNA sequencing details, data filtering and processing protocol are required, that a good practice for every metagenomic study.

L116: what the level (value) of the higher/lower wrinkle grade?

L151: how many samples were sequenced? Please provide the sequencing statistics

L162, L180: the usage of old versions of software (MetaPhlAn ver. 2 instead of current ver. 4; HUMAnN ver. 2 instead of ver. 3) should be explained. The reference databases of these tools were greatly enriched in past years.

L165: NMDS fitting parameters (or software) are unknown

L184: What particular version and date (timestamp) of UniRef was used?

L186: which organisms? Please describe raw data filtering step in details

Some minor corrections to the text (style and spelling):

L89: “wash-out” remove word brake

L157: “abbduk” → “bbduk”; the correct reference to bbduk is main website https://sourceforge.net/projects/bbmap/

Author Response

1. Summary

Thank you very much for taking the time to review this manuscript and making valuable comments and suggestions that will improve the manuscript value and comprehension.  Please find the detailed responses below and the corresponding revisions/corrections highlighted/in track changes in the re-submitted files.

2. Questions for General Evaluation

Reviewer’s Evaluation

Response and Revisions

Does the introduction provide sufficient background and include all relevant references?

Must be improved

We updated the background of this study by adding the most recent suggested references and others.

Are all the cited references relevant to the research?

Can be improved

We’ve updated the bibliographic reference and removed the one which was not relevant and adding new more relevant.

Is the research design appropriate?

Must be improved

We’ve tried to improve by completing with some information regarding the design.  

Are the methods adequately described?

Must be improved

We’ve proceeded to some changes according to reviewers’ comments and suggestions.

Are the results clearly presented?

Must be improved

We’ve proceeded to some changes according to reviewers’ comments and suggestions.

Are the conclusions supported by the results?

Can be improved

We’ve proceeded to some changes according to reviewers’ comments and suggestions.

3. Point-by-point response to Comments and Suggestions for Authors

Comments 1: Section 3.1.3 is unclear at whole for non-specialists. Table 2 has no references in the main text, what the units of measured metadata and how the data were used to build a model? The description in 2.5.2 is very short, it could be clearer with the scheme of data analysis.

Response 1: Thank you for helping us to make this clearer. The purpose of this metadata analysis was to determine which metadata could potentially be correlated with each other. If 2 metadata are correlated, only one of two is introduced in the model to avoid biasing the biological driven analysis. The goal was to limit model terms to just those that were descriptive on their own, e.g. only BMI was included in the model, the term "weight" was then dropped.

We therefore add that in the text in the goal of this analysis Lines 218-221 and we also added a scheme line 234.

For the metadata use for the analysis please refers to metadata section in the additional report provided for a better understanding.

Table 2 has been added in text Lines 342 & 346.

Comments 2: Figure 1: most of the text is unreadable, please correct the font or layout.

Response 2: We totally agree with that. We have, accordingly, increased the font size and removed some edits which does not bring some information mentioned in the results and the discussion. Line 296.

Comments 3: L292: “highly correlated factors” – correlated with which variable?

Response 3: BMI was highly correlated to the Weight and shampoo frequency with conditioner frequency Line 343.

Comments 4: L375: reference [36] devoted to the skin disease, not age-related

Response 4: Yes, the O’Neil’s reference is linked to acne; we removed it and replaced by the relevant one from Dimitriu et al. line 442, that is now referenced as n°33 .

Comments 5: L379: here authors talk about bacterial species (13 of 19), not microorganisms?

Response 5. Microorganism replaced by bacteria Line 444.

Comments 6: First question – the raw sequencing data are missing, so the results cannot be reproduced. The DNA sequencing details, data filtering and processing protocol are required, that a good practice for every metagenomic study.

Response 6: We understand this request. The sequencing was outsourced and to access the raw sequencing data we need to have access to some archives; and to make these data accessible to scientific community, this would take sometimes. Nevertheless, we can supply a more detailed report containing protocols and input data that we use for the microbiota analysis as well as some data on prevalence. This report would help also as it mentioned for each analysis and model the input data regarding DNA sequencing, filtering steps and processing protocol.

Comments 7: L116: what the level (value) of the higher/lower wrinkle grade?

Response 7: Thank you for asking. We added the information Lines 128 then 145-146.

Comments 8: L151: how many samples were sequenced? Please provide the sequencing statistics.

Response 8: 242 samples were sequenced over 288 samples Line 182. The average sequencing depth was 9.9 Gbase per sample.

Comments 9: L162, L180: the usage of old versions of software (MetaPhlAn ver. 2 instead of current ver. 4; HUMAnN ver. 2 instead of ver. 3) should be explained. The reference databases of these tools were greatly enriched in past years.

Response 9: This study was done between 2017 and 2018, so the data provided in this study are those obtained with these old versions. Humann2 downloads the database itself. The version of Humann2 that was ran was v0.11.1 and it was ran once in October of 2017 and again February of 2018. Human2 v0.11.1 came out April of 2017 so the database would have been downloaded sometime between April 2017 and October 2017. Added in the text Lines 228-244.

Comments 10: L165: NMDS fitting parameters (or software) are unknown

Response 10: Bray Curtis dissimilarity matrix derived from the microbiome composition was used to construct clusters of samples. The non-metric multidimensional scaling (NMDS) was used to evaluate the sample structure. Statistical significance of association between microbiome composition and cohort was determined using one-way/tow-way ANOVA for multiple comparisons, the t-test, and the Chi-Square test, as appropriate; differences P < 0.05 were noted as significant. False discovery rate (FDR) was also reported with Benjamini–Hochberg procedure. Spatial distribution of the species having significant changes in the relative abundance between the cohort in the different skin sites was generated. All analysis was performed in R with different packages. Added in the text Lines 194-254.

Comments 11: L184: What particular version and date (timestamp) of UniRef was used

Response 11: UniRef90 database was used. Humann2 downloads the database itself. The version of Humann2 that was ran was v0.11.1, once in October of 2017 and again February of 2018. Human2 v0.11.1 came out April of 2017, so the database would have been downloaded between April 2017 and October 2017. Lines 238-244.

Comments 12: L186: which organisms? Please describe raw data filtering step in details

Response 12: Thank you for bringing this up. A filtering step was conducted to remove bacteria; Gene Ontology with less than a 25% prevalence in the total dataset. A threshold was set up also to remove species that show very low abundance across samples; this threshold is typically set close to what we consider our detection threshold 10-6.  Addition in the text. Lines 222-227.

Comments 13: L89: “wash-out” remove word brake

Response 13: Done Line 118.

Comments 14: L157: “abbduk” → “bbduk”; the correct reference to bbduk is main website https://sourceforge.net/projects/bbmap/

Response 14: Done. Lines188-189.

4. Response to Comments on the Quality of English Language

Point 1: No major comment

5. Additional clarifications: we have no additional clarification.

Round 2

Reviewer 1 Report

Comments and Suggestions for Authors

The authors have revised the manuscript according to the suggestions; the manuscript is improved and can be published.

Comments on the Quality of English Language

minor  editing needed

Author Response

Thank you

Reviewer 2 Report

Comments and Suggestions for Authors

I would to thank authors for the improving figures and the manuscript, but raw data availability remains unclear. 

It takes little time to publish raw data to NCBI GenBank (several days on my experience), and I strongly recommend to do this. Raw data make your paper more attractive (so for citing).

Author Response

Our data has been finally published in Bioproject under the reference

PRJNA1099173
